# Regenerative Medicine Applied to Musculoskeletal Diseases in Equines: A Systematic Review

**DOI:** 10.3390/vetsci10120666

**Published:** 2023-11-23

**Authors:** Andrea Pérez Fraile, Elsa González-Cubero, Susana Martínez-Flórez, Elías R. Olivera, Vega Villar-Suárez

**Affiliations:** 1Department of Anatomy, Veterinary Faculty, Campus de Vegazana, Universidad de León, 24007 León, Spain; 2Institute of Biomedicine (IBIOMED), Veterinary Faculty, Campus de Vegazana, Universidad de León, 24007 León, Spain; 3Department of Molecular Biology, Veterinary Faculty, Campus de Vegazana, Universidad de León, 24007 León, Spain

**Keywords:** horse, osteoarthritis, musculoskeletal disease, regenerative therapy, mesenchymal stem cells

## Abstract

**Simple Summary:**

Musculoskeletal injuries have significant importance within the field of equine health, mainly due to their economic and sporting implications in today’s world. The tissues most affected by such injuries in horses include tendons, ligaments, and cartilage, all of which have a restricted ability to self-heal. This is where regenerative therapies arise, which involve the use of living cells and non-cell therapies to treat various diseases, injuries, and medical conditions and to restore the affected tissue to its native state, both structurally and functionally. Platelet-rich plasma and autologous conditioned serum are widely used in regenerative medicine. In the last few years, mesenchymal stem cells, a type of adult stem cell found in various tissues throughout the body, including bone marrow, adipose (fat) tissue, and the umbilical cord, have attracted attention for their therapeutic potential. These mesenchymal stem cells (MSCs) can differentiate into various types of tissues, making them versatile agents for tissue regeneration. In addition, they have anti-inflammatory properties, crucial for mitigating the harmful effects of inflammation often associated with injuries, and immunomodulatory capabilities, which can help regulate the immune system response. Cell therapy therefore harnesses the power of MSCs to combat the challenges posed by musculoskeletal injuries in horses, offering hope for faster recovery and a return to peak performance.

**Abstract:**

Musculoskeletal injuries in horses have a great economic impact, predominantly affecting tendons, ligaments, and cartilage, which have limited natural regeneration. Cell therapy, which uses mesenchymal stem cells due to their tissue differentiation properties and anti-inflammatory and immunoregulatory effects, aims to restore damaged tissue. In this manuscript, we performed a systematic review using the Parsifal tool, searching the PubMed and Web of Science databases for articles on regenerative medicine for equine musculoskeletal injuries. Our review covers 17 experimental clinical studies categorized by the therapeutic approach used: platelet-rich plasma, conditioned autologous serum, mesenchymal stem cells, and secretome. These therapies reduce healing time, promote regeneration of fibrocartilaginous tissue, improve cellular organization, and improve joint functionality and sustainability. In conclusion, regenerative therapies using platelet-rich plasma, conditioned autologous serum, equine mesenchymal stem cells, and the emerging field of the secretome represent a promising and highly effective approach for the treatment of joint pathologies in horses, implying a valuable advance in equine healthcare.

## 1. Introduction

Among all musculoskeletal diseases present in horses, joint problems rank first among the most common causes of lameness, followed by tendon injuries as the main cause of reduced performance. The type and anatomical location of musculoskeletal pathologies vary between different athletic disciplines, competition levels, and the age of the animal [1]. These joint diseases can be caused by traumas, infections, genetic factors, or malformations, and they are all encompassed under the term “arthropathies”. Common symptoms of these diseases include pain, inflammation, stiffness, and lameness in the affected joints. Articular pathologies can affect one or multiple components of the joint, including the joint membranes, surrounding tendons, underlying bone, cartilage, bursae, and synovial fluid within the joint [2]. Their main limitation is the limited regenerative capacity [3,4]. The appearance of lesions in any of the tissues or organs involved in the musculoskeletal system of horses will result in the animal’s inability to perform its usual activities, whether it be basic work or participation in high-level competitions [5]. The problem arises when the initial cause is not diagnosed and treated promptly, leading to more significant issues such as osteoarthritis (OA), characterized as a progressive and permanent disease affecting the joint’s articular cartilage, bone tissues, and associated soft tissues [6,7,8]. Secondly, arthritis is a disease that affects the joints and is characterized by their inflammation. It causes pain, stiffness, swelling, and difficulty moving the affected joints. There are different types of arthritis, such as traumatic arthritis and septic arthritis (SA) [9]. Additionally, the inflammation of the tendon or peritendinous tissue, such as tendinitis or tenosynovitis, is of special interest. Tendinitis is the inflammation of a tendon, which may predispose horses to future fiber ruptures with associated hemorrhages and oedemas. It is more common in horses performing high-impact and high-speed work, such as racehorses, as factors like excessive fatigue, tendon overextension, poor physical conditioning, poor racetrack conditions, improper shoeing, or joint malformations can predispose them to tendinitis [9]. Finally, tenosynovitis is the inflammation of the tendon sheath, which consists of a synovial membrane and a fibrous layer. In cases of tenosynovitis, both parts of the tendon sheath are inflamed, leading to increased synovial fluid and distension of the tendon sheath [10].

### 1.1. Conventional Treatment of the Most Common Musculoskeletal Pathologies in Equine Medicine

The primary goals of treating osteoarticular pathologies include pain control, maximizing treatment function, limiting disease progression, and, theoretically, facilitating the repair processes [11]. Most often, a combination of management measures, medication, and, in some cases, surgical procedures are involved. The treatment approach may vary depending on the underlying cause, severity of the condition, and the horse’s general health. The main and most used treatment for pain management is non-steroidal anti-inflammatory drugs (NSAIDs). In addition to controlling pain by inhibiting prostaglandin release, they are believed to have central actions unrelated to cyclooxygenase. Phenylbutazone is the most used NSAID for managing musculoskeletal inflammations and pain in horses, particularly in OA [12,13]. However, NSAIDs are considered only moderately effective in controlling the pain response to specific pathologies such as OA, although their efficacy improves when combined with other therapies mentioned below [14]. Intra-articular corticosteroids are the most widely used drugs for intra-articular treatment of non-infectious synovitis and OA, thanks to their potent anti-inflammatory effect and indirect analgesic activity [15]. In cases where these treatments do not yield results, surgical treatments such as joint lavage, arthroscopy for joint problems, or tenotomy/desmotomy for soft tissue injuries are available. Three decades ago, due to the limitations of conventional treatments, the application of regenerative medicine in veterinary medicine emerged [16]. Complementary treatments can also be applied to achieve better treatment efficacy. Techniques such as shoeing aimed at absorbing impact on the ground, acupuncture, electromagnetic stimulation therapies, muscle stimulation through electric impulses, laser therapy, etc., have shown their usefulness in numerous studies regarding the clinical resolution of arthritis in horses [14].

### 1.2. Regenerative Medicine

Regenerative medicine (RM) is a rapidly developing discipline in current medicine, focused on the restoration and functional regeneration of tissues and organs during severe injuries or chronic conditions [17]. The goal of RM is to restore or improve normal body function using therapies with a biological basis [18]. RM encompasses fields of research such as tissue engineering (TE) or the stimulation of endogenous mechanisms for the self-repair of damaged tissues. In recent years, orthobiology has been increasingly used in the treatment of equine musculoskeletal diseases, so understanding how these new treatments modulate joint inflammatory responses from a mechanistic perspective is crucial [19]. In equine practice, several modalities of regenerative therapies are used, including the frequent use of mesenchymal stem cells (MSCs), platelet-rich plasma (PRP), interleukin-1 receptor antagonist protein (IRAP), conditioned autologous serum (ACS), and autologous protein solution (APS), among others [20]. Currently, MSC therapy, IRAP-induced ACS, and PRP are the three main types of regenerative medicine (Figure 1) used in equine musculoskeletal injuries [18].

#### 1.2.1. Platelet-Rich Plasma (PRP)

Platelets (also known as thrombocytes) are small cytoplasmic fragments without a nucleus derived from cells produced in the bone marrow (BM), the megakaryocytes, which circulate in the bloodstream and aid in blood clotting. Platelets are a source of various growth factors (GF) and other proteins that function in tissue repair stimulation, inflammation reduction, induction of chemotaxis in mesenchymal cells, and promoting cell proliferation and differentiation [21]. The main interest in PRP lies in the growth factors released by platelets. These include transforming growth factors β1 and β2 (TGF-β1 and TGF-β2), platelet-derived growth factors (PDGF), insulin growth factor (IGF-I), epidermal growth factor (EGF), and hepatocyte growth factor (HGF), among others [22]. Technically, PRP is considered a plasma concentrate with platelets and leukocytes, and it is estimated to contain between 300 and 400 × 10^3^ platelets per mL of plasma, with a similar leukocyte concentration [23]. However, there is still limited information on the complex molecular mechanism by which PRP acts positively in patients with joint diseases. Nevertheless, as mentioned earlier, the GF contained in PRP (such as HGF, PDGF, TGF-β1, and IGF-I) have beneficial effects on effected cartilage and synovial membrane. Additionally, these GF have anti-inflammatory properties and can increase the vascularization of these tissues, leading to an increase in the metabolic rate of the constituent cells [24].

#### 1.2.2. Autologous Conditioned Serum (ACS)

Platelet-rich plasma (PRP) and autologous conditioned serum (ACS) were among the initial orthobiologic treatments introduced for equine patients. ACS is a biological product used in regenerative medicine and the treatment of various medical conditions which is obtained from the patient’s blood. The collection process is the same as that used for PRP. In the case of ACS, the blood sample is also centrifuged to obtain plasma but then undergoes an additional activation process by adding substances such as calcium chloride. This stimulates the release of growth factors and other bioactive components present in platelets [25]. The therapy with an autologous serum enriched in interleukin-1 (IL-1Ra), known as IRAP (interleukin-1 receptor antagonist protein), is a therapeutic approach used in the treatment of osteoarthritis in horses. Although IRAP has been primarily used in veterinary medicine, its potential has also been explored in human medicine [26]. Interleukin-1 receptor antagonist protein (IRAP) is an endogenous protein mainly produced by cells, primarily monocytes. It is a competitive antagonist of interleukin-1 (IL-1) whose main function is to act as a mediator of inflammation and degradation in joints. The final product of this process is called ACS as it contains other cytokines and GF in addition to IRAP [18]. In horses, the use of ACS is recommended for inflamed joints, synovitis, capsulitis, and OA. For now, ACS is only recommended for intra-articular administration, as its effect on tendons, ligaments, bursae, and tendon sheaths is still unknown. Despite this lack of knowledge, Frisbie et al., 2007 [25], have shown better clinical outcomes in horses with experimentally induced OA treated with ACS [5,26].

#### 1.2.3. Autologous Protein Solution (APS)

Recently, a novel patient-side product has been introduced, aiming to combine the advantages of ACS and PRP. Autologous protein solution (APS) concentrates platelets, growth factors, and anti-inflammatory cytokines. Unlike ACS, the production of APS does not necessitate a 24-hour incubation period, making it a more attractive option for equine veterinarians [27].

#### 1.2.4. Mesenchymal Stem Cells (MSCs)

Mesenchymal stem cells are a type of stem cell that can differentiate into various cell types, including bone cells, cartilage cells, and adipose cells. Their main function lies in the regeneration of damaged, diseased, or aged tissues. MSCs are being investigated for their possible use in regenerative medicine, making them a feasible option for cell-based therapies [18]. Their capacity for rapid and indefinite self-replication, as well as their potential to differentiate into various specialized cell types, makes them a key tool for regenerative medicine and cellular therapy for tissue and organ impairments of varying degrees [28].

In horses, MSCs have been isolated from numerous sources, including bone marrow (BM), subcutaneous adipose tissue (AT), peripheral blood, synovial membrane and fluid, amniotic membrane, and umbilical cord blood or tissue [29]. However, despite the wide variety of possible sources for obtaining MSCs, not all are equally effective for clinical purposes. Currently, the most used tissue sources for MSC isolation in equine medicine are BM and subcutaneous AT. However, recent studies have demonstrated the usefulness of using peripheral and perinatal blood for MSC isolation due to the relative ease of obtaining these tissues [30]. Nevertheless, comparative studies between MSC isolation techniques did not show significant differences in the morphology and immunophenotype of adipose tissue-derived MSCs (ATMSCs) compared to BM-derived MSCs (BMMSCs) and umbilical cord-derived mesenchymal stem cells (UCMSCs) [29]. The first evidence of using MSC therapy in vivo in horses involved the reimplantation of autologous cultured expanded BMMSCs into a central lesion of the superficial digital flexor tendon (SDFT), which is one of the most important causes of lameness in horses. This research highlighted the feasibility of using cultured expanded MSCs for therapeutic purposes and showed no adverse reactions 10 days and even 6 weeks after injection [31].

Regeneration of the medial meniscus was demonstrated, leading to subsequent improvement in OA. Since then, MSCs have been successfully used for the treatment of intra-articular soft tissue injuries in horses [32], as well as for cartilage regeneration. However, there have been variable results when using them for primary OA [33]. As with other cell therapies, many sources of variation need to be considered before treating patients with intra-articular MSCs. These factors include the source of stem cells, collection and proliferation techniques, shipping, and transportation effects, as well as the vehicle and needle size to be used for injection [6].

#### 1.2.5. MSCs-Derived Secretome

Initially, it was believed that the therapeutic effects of transplanted MSCs were mediated by MSCs which migrated to the damaged region and differentiated into the involved cells. However, recent research has suggested that MSCs secrete a wide range of trophic factors to achieve their therapeutic effects [34]. The secretome of stem cells, also known as conditioned medium (CM), contains paracrine-soluble substances and vesicles that MSCs can produce. Studies have shown that the secretome of adipose tissue-derived MSCs contains growth factors, cytokines, chemokines, and extracellular vesicles, which play crucial roles in modulating cellular responses and promoting tissue healing. These proteins are believed to be responsible for regulating angiogenesis, immune response, tissue protection, and wound healing. As a result, there would be significant benefits if the therapeutic effects based on MSCs were replaced by trophic factors obtained from MSCs [35].

The main aim of this work is to conduct a systematic literature review of regenerative medicine therapies applied to musculoskeletal pathologies in horses.

## 2. Materials and Methods

To achieve the aim of this work, a systematic review was carried out using the Parsifal search and article selection tool. This systematic review was conducted following the PRISMA (Preferred Reporting Items for Systematic Reviews and Meta-Analyses) statement, an evidence-based guideline for systematic reviews [36,37].

### 2.1. Search Strategy

An exhaustive search strategy for articles related to regenerative medicine applied to musculoskeletal injuries in horses was performed. The search was conducted in the period between November 2022 and April 2023.

#### 2.1.1. PICOC Question

To establish the planning, the following PICOC question was formulated:Population: horses with musculoskeletal pathologies;Intervention: regenerative medicine or cell therapy;Comparison: conventional therapy in musculoskeletal pathologies;Outcome: regenerative and anti-inflammatory effects;Context: academic.

#### 2.1.2. Search Questions

The selection of articles was carried out by seeking answers to a series of questions that were previously formulated. These questions were as follows:Is regenerative medicine based on cell therapy effective in horses?What types of therapies are most used in regenerative medicine in horses?What types of pathologies can be treated with regenerative medicine in horses?When is regenerative medicine the best option for treating musculoskeletal pathologies?Is the secretome of MSCs used in horses?

#### 2.1.3. Search String

The following search string was used for searching: (“Horse*” OR “Equine”) AND (“Musculoskeletal disease*” OR “Osteoarthritis” OR “Arthritis”) AND (“Regenerative medicine” OR “Secretome” OR “Conditioned medium” OR “Stem cell*”) AND (“Immunomodulation” OR “Regeneration”).

#### 2.1.4. Databases

The articles here analyzed were obtained from the following databases:COCHRANE (https://www.cochranelibrary.com/); accessed on 1 January 2022.PUBMED (https://pubmed.ncbi.nlm.nih.gov/); accessed on 15 January 2022.Science Direct (http://www.sciencedirect.com); accessed on 15 January 2022.Web of Science (http://wos.fecyt.es/); accessed on 1 February 2022.Scopus (http://scopus.fecyt.es/); accessed on 2 February 2022.Springer Link (http://link.springer.com); accessed on 21 January 2022.Wiley Online Library (https://onlinelibrary.wiley.com/) accessed on 19 February 2022.

### 2.2. Eligibility Criteria

The eligibility criteria were established before the article search to include them in the review. Therefore, the following criteria were selected for inclusion:Articles in Spanish and English;Full text availability;Clinical trials;Published within the last 10 years.

On the other hand, exclusion criteria were based on the following:Other pathologies unrelated to osteoarticular issues;Studies made on other species different from horses, including human beings;No use of regenerative medicine.

## 3. Results and Discussion

This work includes articles published from 2013 to 2023. However, due to the limited knowledge of the subject, it was considered appropriate to include four articles from previous years. In the initial search, 372 articles were selected from different sources (COCHRANE: 0; PUBMED: 41; Science Direct: 96; Scopus: 36; Springer Link: 100; Web of Science: 44; Wiley Online Library: 55). After eliminating 59 duplicate articles, the exclusion criteria were applied, and a thorough review of the title and abstract of each article was conducted. Finally, 17 articles were selected for the literature review. Figure 2 shows the percentage of selected works in each database.

The data extraction process for this review complied with the inclusion criteria established at the outset. Figure 3 shows the articles which were finally included in the study.

A flow chart (Figure 4) summarizes the study selection process. After data extraction, 372 articles were obtained, 59 duplicate articles were eliminated, and after applying the exclusion criteria and reviewing the documents by author, summary, and relationship with the topic to be treated, 296 articles were eliminated due to their remoteness from the study, leaving 17 articles to be selected for the literature review.

The 17 experimental studies were grouped into four tables. Table 1, comprising five studies which deal with the application and results of PRP in musculoskeletal pathologies; Table 2, comprising four studies which deal with the application of ACS in osteoarticular injuries; Table 3, in which six studies that use MSCs are grouped; and finally, Table 4, comprising two articles which deal with the action of the secretome in the injuries that have been dealt with in this review.

PRP is commonly used to treat tendon and ligament injuries in horses due to its ease of administration through ultrasound-guided injections at central lesions. It is particularly effective when used in the acute phase, characterized by hypoechoic areas visible on ultrasound. The preparation of PRP is performed in situ with the horse’s blood and is then used to fill the defects in the injured tissue [38].

**Table 1 vetsci-10-00666-t001:** PRP application in osteoarticular lesions in equine patients.

References	Study Design	Number of Equine Patients	Therapy	Lesion Treated
Torricelli et al., 2011 [3]	Controlled experimental study	13	PRP associated with bone marrow mononucleated cells	Lameness
Scala et al., 2014 [39]	Controlled experimental study	150	No-gelled platelet concentrate	Teno-desmic injuries
Yamada et al., 2016 [40]	Controlled experimental study	12	PRP gel associated with stem cells from adipose tissue	Chondral defects
Geburek et al., 2016 [24]	Controlled experimental study	10	PRP concentrate	Tendinopathy of the superficial digital flexor tendon
Beerts et al., 2017 [41]	Controlled experimental study	104	PRP and allogenic tenogenically induced mesenchymal stem cells	Tendon and ligament injuries

**Table 2 vetsci-10-00666-t002:** ASC application in osteoarticular lesions in equine patients.

References	Study Design	Number of Equine Patients	Therapy	Lesion Treated
Frisbie et al., 2007 [25]	Controlled experimental study	16	ACS	Osteoarthritis
Georg et al., 2010 [42]	Controlled experimental study	7	Autologous conditioned plasma (ACP)	Severe tendinitis of the superficial digital flexor tendon, deep digital flexor tendon, or desmitis of the inferior check ligament
Geburek et al., 2015 [43]	Controlled experimental study	15	ACS	Tendinitis of the superficial digital flexor tendon
Marques-Smith et al., 2020 [44]	Controlled experimental study	20	ACS	Articular lameness

**Table 3 vetsci-10-00666-t003:** MSCs application in osteoarticular lesions in equine patients.

References	Study Design	Number of Equine Patients	Therapy	Lesion Treated
Mokbel et al., 2011 [45]	Controlled experimental study	27 donkeys	BMSCs	Osteoarthritis
Carvalho et al., 2013 [46]	Controlled experimental study	8	ASCs and PRP concentrates	Tendinitis of the superficial digital flexor tendon
Renzi et al., 2013 [47]	Clinical trial	33	BMSCs	Tendonitis and desmitis
Broeckx et al., 2014 [48]	Controlled experimental study	20	PRP and peripheral blood mononuclear cells (PBMCs)	Degenerative joint disease
González-Fernández et al., 2016 [49]	Controlled randomized and experimental study	6	ASCs and BMSCs	Meniscal defects
Villatoro et al., 2018 [50]	Clinical trial	80	ASCs	Osteoarthritis

In the article by Torricelli et al., 2011 [3], the main objective was to evaluate the clinical benefit of administering a combination of autologous PRP and freshly isolated bone marrow mononuclear cells (BMMNCs) in competition horses effected by musculoskeletal overload injuries. However, some assumptions expressed by them could encourage the use of this combined biological therapy without sufficient scientific basis. This view is mainly based on the authors not including a control group or at least a group that received only PRP. It is irresponsible to recommend the use of this combined biological treatment without an appropriate study design. In conclusion, the authors of this article believe that further research with an appropriate study design including control groups is required to determine more accurately the efficacy and role of BMMNCs compared to PRP in treating musculoskeletal injuries in competition horses.

Scala et al., 2014 [39], aimed to evaluate the safety and clinical outcome of PRP treatment in tendinous injuries and desmitis in competition horses. They were able to demonstrate that treatment with platelet-derived growth factors leads to the formation of a tendon with normal morphology and functionality, resulting in the resumption of agonistic activity in the treated horses.

In their study, Yamada et al., 2016 [40], investigated the use of activated PRP as a fibrin gel scaffold in combination with MSCs for the treatment of chondral defects in horses. They concluded that the joints treated with PRP gel and MSCs showed improved macroscopic and microscopic appearance in tissue repair compared to the control group. These results suggest that PRP gel could be an effective option in chondral defect repair and support for MSC implantation.

For Geburek et al., 2016 [24], the purpose of their placebo-controlled clinical trial was to describe the effect of a single PRP treatment on SDFT injuries in horses within clinical and ultrasound parameters. A single intralesional treatment with PRP up to 8 weeks after the onset of clinical signs of tendinopathy contributes to an earlier reduction in lameness compared to treatment with saline solution and advanced organization of the repair tissue, as the fibrillar matrix is organized into fascicles while remodeling continues.

In the following study, Beerts et al., 2017 [41], performed an intralesional injection with tenogenically induced MSCs and PRP 5 to 6 days after the diagnosis of a suspensory ligament (SL) injury (n = 68) or SDFT (n = 36) injury. The findings from their study support the use of regenerative therapies in the treatment of musculoskeletal injuries in sport horses, offering the potential for better recovery and a lower risk of new injuries compared to conventional approaches.

Overall, PRP is a valuable therapeutic option for various musculoskeletal conditions in horses, offering regenerative and anti-inflammatory effects that can aid in the healing process and improve clinical outcomes. Long-term treatment with PRP has the potential to increase the number of horses that return to their previous performance level. Early treatment of tendinopathy with PRP should be considered to enhance these effects [24].

However, further research and studies are required to confirm these findings and determine the clinical viability of this therapy in treating chondral defects [40,41]. However, careful evaluation of potential adverse effects and cost–benefit considerations are also necessary before recommending this therapeutic approach in clinical practice [3].

Biological therapies, such as ACS, are gaining popularity in the treatment of orthopedic conditions in equine veterinary medicine due to their remarkable potential for promoting natural healing processes and enhancing the overall well-being of equine patients. 

Frisbie et al., 2007 [25], to evaluate the effects of intra-articular administration of ACS in the treatment of experimentally induced OA in horses, induced OA in a middle carpal joint of 16 horses. The results not only showed no adverse events related to the treatment but also that the horses treated with ACS exhibited significant clinical improvement. Based on these findings, the study concluded that ACS treatment led to significant clinical and histological improvement in OA-effected joints in horses compared to placebo treatment. The results suggest the potential effectiveness of ACS in treating OA in horses.

In this case, Georg et al., 2010 [42], sought the efficacy of the same treatment in severe tendinitis of the SDFT and deep digital flexor tendon (DDFT) or desmitis of the inferior check ligament in seven horses; again, the treatment results showed positive outcomes in all horses during the follow-up period. All horses returned to the same workload they had previously undertaken or were back in full training. The use of ACS appeared to be effective in promoting healing and recovery in these tendon and ligament injuries.

Geburek et al., in 2015 [43], investigated the effects of a single intralesional injection of ACS in naturally occurring tendinopathies in the SDFT of horses. The results showed a significant decrease in lameness, swelling, and tendon injury in the limbs treated with ACS. In conclusion, a single intralesional injection of ACS in the SDFT of horses with acute tendinopathy resulted in an early and significant reduction in lameness and temporary improvement in ultrasound parameters of the repair tissue. ACS treatment reduced tenocyte proliferation and increased their differentiation, as evidenced by the high expression of type I collagen during the remodeling phase.

Lastly, regarding the association between cytokine and growth factor content in ACS, Marques-Smith et al., in 2020 [44], proposed to study this synergy and its clinical effect in racehorses with mild joint lameness. It was demonstrated that the therapeutic benefits of ACS may be related to higher levels of IL-1Ra and IGF-1. This study supports previous findings of significant variability between individuals in cytokine and growth factor content in ACS.

Despite the overall success of the studies, authors like Frisbie et al., 2007 [25], and Geburek et al., in a study published in 2015 [43], recommend conducting further controlled clinical trials to evaluate this treatment, in addition to providing stronger evidence and supporting these clinical observations in the medium and long term [42].

The use of MSCs for cell therapies is based on their ability to target and integrate into the target tissue in the long term. MSC therapy has been applied in bone and cartilage repair, as well as in the treatment of osteoarthritis [45].

In their study, Mokbel et al., 2011 [45], investigated the evidence of recruitment and the repairing effect of MSCs in the healing process of experimentally induced OA in donkeys, demonstrating the repairing effect of MSCs. They could confirm the recruitment ability of MSCs by incorporating injected and GFP-labeled MSCs into the newly formed and repaired cartilage.

Carvalho et al., 2013 [46], opted for treatment through intralesional administration of ATMSCs suspended in platelet concentrate (PC) and buffered phosphate saline (PBS), respectively. Better results were reported in the group treated with ATMSCs suspended in PC. The use of ATMSCs combined with PC for the treatment of experimentally induced tendinitis prevented tendon lesion progression and resulted in increased organization and decreased inflammation. These data demonstrate the therapeutic potential of this therapy for the treatment of tendinitis in horses.

The clinical trial developed by Renzi et al., 2013 [47], demonstrated both the efficacy of treating horses with bone marrow-derived MSCs and their ability to resume their prior level of athletic engagement.

Broeckx et al., 2014 [48], used undifferentiated MSCs after chondrogenic induction, injecting the following for their study: (1) PRP; (2) MSCs; (3) MSCs and PRP; or (4) chondrogenically induced MSCs and PRP. The combined use of PRP and MSCs significantly improved the functionality and sustainability of damaged joints compared to PRP treatment alone. However, higher ranges of short-term clinical improvement were obtained with chondrogenically induced MSCs and PRP.

On the other hand, González-Fernández et al., 2016 [49], focused on evaluating the regenerative capacity of BMMSCs or ATMSCs in horse meniscal lesions using a collagen biomaterial to carry the MSCs in six mares. One year after the experiment, they observed that the treated defects regenerated with fibrocartilaginous tissue, while the untreated defects were partially repaired or not repaired. Thus, the study concludes with evidence that AT-derived MSCs could be a good alternative to BM-derived MSCs for regenerative treatments, especially in meniscal lesions.

Finally, and addressing not only the effectiveness but also the safety of this treatment, Villatoro et al., 2018 [50], aimed to evaluate the effectiveness and safety of repeated administrations of allogeneic ASCs in horses with OA clinical signs. Clinical and synovial inflammatory signs were reduced more rapidly in the groups treated with MSCs, and the repeated allogeneic administration did not produce adverse reactions.

In this case and based on a review of pre-clinical studies in animals, Xing et al., 2018 [51], sought to determine the evidence supporting the use of MSCs in future clinical trials for knee osteoarthritis (KOA). According to the available evidence from pre-clinical animal studies, the injection of MSCs in clinical trials for KOA cannot be reliably recommended. The included studies presented a high risk of bias, and the quality of evidence for all outcomes was low or very low. More high-quality experimental studies and efforts are needed to effectively translate the findings of pre-clinical studies into clinical trials to determine the potential of MSCs in treating KOA.

The significant recovery demonstrated in most of the studies confirms that intra-articular injection of MSCs, whether autologous or chondrogenically induced [47], is a viable and practical option to treat different degrees of OA [45]. It can be observed that the progression of the lesion is prevented, and there is greater organization of collagen fibers and a reduction in inflammatory infiltration [46]. As for diagnostic methods, radiology and MRI do not show significant differences in post-treatment joint changes in most cases, while ultrasound can suggest alterations of synovial effusions [52].

**Table 4 vetsci-10-00666-t004:** MSCs-derived secretome application in osteoarticular lesions in horses.

References	Study Design	Number of Horses	Therapy	Lesion Treated
Arévalo-Turrubiarte et al., 2022 [53]	In vitro experiments with randomized complete block design	-	BMSC, ASC, and synovial fluid-derived secretome	Inflammation model of osteoarthritis
Kearney et al., 2022 [34]	Randomized positively and negatively controlled experimental study	8	BMSCs-derived secretome	Joint inflammation

The secretome of MSCs contains soluble factors such as cytokines and growth factors, as well as EVs and exosomes, which contain bioactive molecules like proteins, lipids, and nucleic acids. These secreted factors can modulate immune responses, reduce inflammation, and promote tissue repair and regeneration [53].

In an in vitro experimental study, Arévalo-Turrubiarte et al., 2022 [53], characterized extracellular vesicles (EVs) secreted by equine MSCs to evaluate their impact on equine chondrocytes treated with pro-inflammatory cytokines. They cultured MSCs derived from BM, AT, and synovial fluid. The EVs derived from equine BM–MSCs showed the ability to reduce the gene expression of metalloproteinase-13 in inflamed chondrocytes in vitro. These findings suggest that MSC-derived EVs could reduce inflammation and be considered a complementary treatment to enhance tissue and cartilage repair in joint pathologies.

On the other hand, in this experimental study, Kearney et al., 2022 [34], investigated the anti-inflammatory effect of intra-articular allogeneic MSC secretome in the joints of eight horses with induced inflammation. Allogeneic equine MSC secretome was injected into an assigned joint, and its effect was compared with that of equine MSCs. The results showed that the group treated with the MSC secretome had a smaller joint perimeter and higher levels of GAG in synovial fluid compared to the control group at certain periods, even though no significant differences were found between both treatments in the following phase. It is concluded that intra-articular injection of MSC secretome had clinical anti-inflammatory effects in this equine joint inflammation model. Additionally, MSC secretome could be a commercially viable alternative to MSC treatment in this context.

Although the precise composition and function of the MSC secretome are still being investigated, they represent a promising therapeutic option for various inflammatory and degenerative conditions. However, more research is needed to fully understand the mechanisms underlying the effects of the MSC secretome and optimize its use in clinical applications [34].

## 4. Limitations of the Study

Experimental studies in horses show promising results in terms of tissue repair and regeneration, largely due to the potential benefits of MSCs. However, further research in this field is considered necessary to support its effectiveness and establish clear guidelines for its use in clinical practice.

The lack of controlled clinical trials limits the strength of this evidence. Moreover, the importance of conducting medium- and long-term evaluations to determine the durability and sustained efficacy of these therapies is highlighted. This is particularly relevant in cases of chronic or degenerative injuries, where long-term evaluation is essential to understand the real impact of the treatment.

The lack of factual conclusions based on long-term analysis, coupled with high economic costs compared to other regenerative medicine techniques, implies the need to perform a cost–benefit analysis to ensure and justify the use of this type of therapy in clinical practice.

## 5. Conclusions

Overall, regenerative medicine based on cell therapy is highly effective in osteoarticular injuries in horses due to the regenerative and anti-inflammatory properties of these therapies, making them a promising alternative for the treatment of musculoskeletal injuries in horses. The most commonly used cellular therapies in regenerative medicine in equine clinics today include mesenchymal stem cell (MSC) therapy, platelet-rich plasma (PRP), autologous conditioned serum (ACS), and other therapies in the developmental phase such as the secretome. The pathologies most treated with regenerative medicine, obtaining satisfactory results, include bone tissue injuries, ligament and tendon injuries, osteoarthritis, and degenerative joint diseases. Regenerative medicine is considered a treatment option when conventional therapies have not yielded satisfactory results or when a more advanced therapeutic option focused on tissue regeneration is sought.

Moreover, it should be pointed out that these results obtained in the treatment of injured horses, in addition to their inherent economic implications, have an enormous interest in translational medicine to human beings in this context. These kinds of therapies seem to be preferable to other potential therapies, i.e., replacement surgery, with its inherent decrease in the fitness of the patient, and avoid the risks and ethical concerns of other potential treatments like gene therapies.

## Figures and Tables

**Figure 1 vetsci-10-00666-f001:**
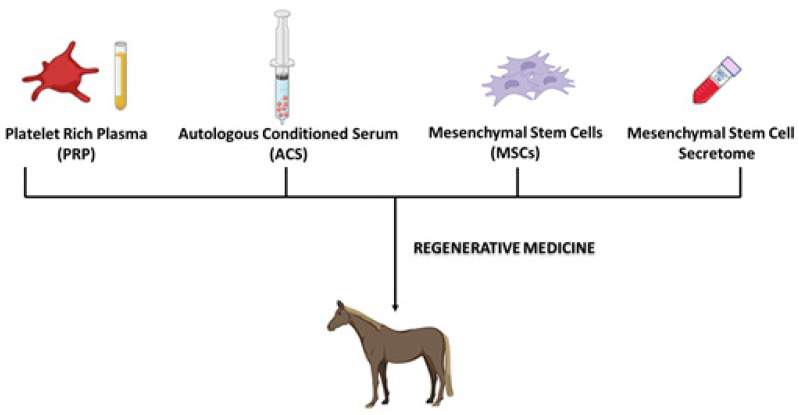
Main modalities of regenerative medicine used in horses.

**Figure 2 vetsci-10-00666-f002:**
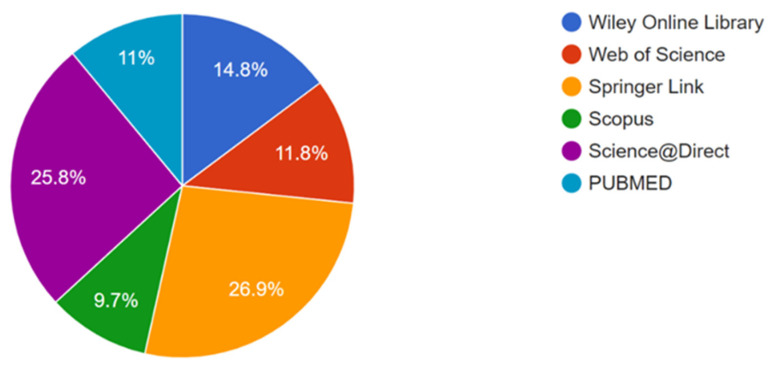
Percentage of articles selected by the database. Source: Parsifal.

**Figure 3 vetsci-10-00666-f003:**
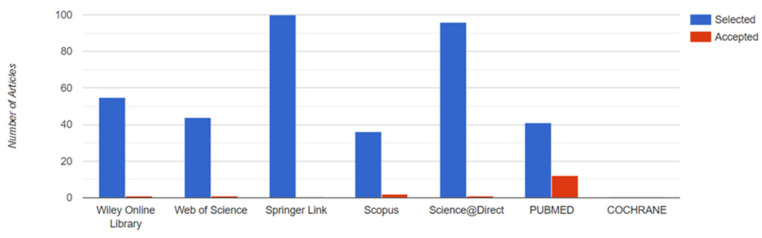
Number of articles accepted by the database. Source: Parsifal.

**Figure 4 vetsci-10-00666-f004:**
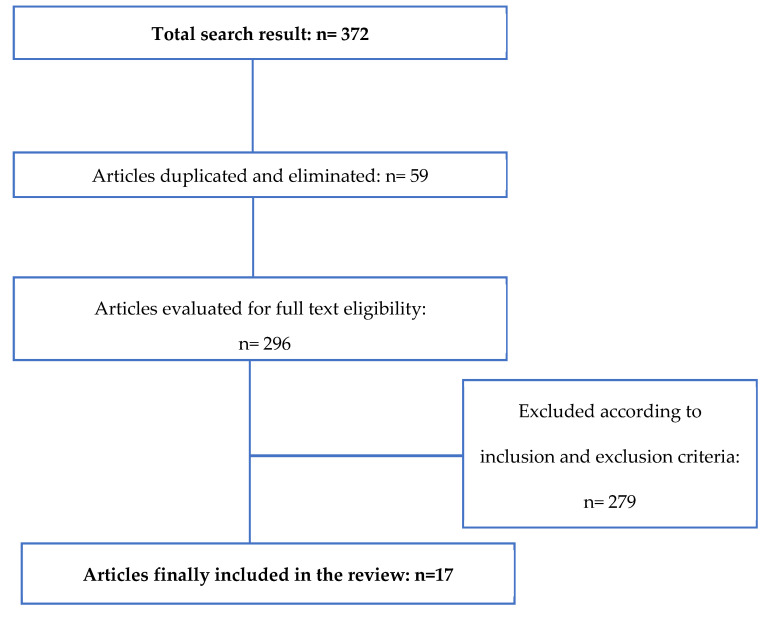
Explanatory flow chart of the item selection process.

## Data Availability

Not applicable.

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
