# Peer review of "Regenerative Medicine Applied to Musculoskeletal Diseases in Equines: A Systematic Review"

_vetsci, 2023, doi:10.3390/vetsci10120666_

Round 1

Reviewer 1 Report

Comments and Suggestions for Authors

The review entitled: “Regenerative medicine applied to osteoarticular diseases in equines: a systematic review” is a well written concise, systematic review that addresses a relevant topic in equine medicine. Before final approval several changes should be performed in the manuscript.

While the title refers to osteoarticular diseases in equines, the search questions were focused on musculoskeletal diseases, which is a broader topic encompassing conditions such as tendinopathies not associated to articular diseases. Therefore, the authors should either change the title to 'musculoskeletal diseases' or restructure the manuscript accordingly.

The simple summary primarily focuses on cell-based therapies and does not encompass all the work, particularly non-cell therapies such as PRP and ACS. Please reformulate this section accordingly.

“1.1. Conventional treatment of the most common osteoarticular pathologies in equine medicine”. Please review this section accordingly with the initial comment on title.

Line 112 and 113: This line suggests that regenerative medicine research started recently, although the results obtained may not yet meet the desired criteria; the search was initiated years ago. Please reformulate this sentence accordingly.

Line 129- While the authors acknowledge autologous protein solution (APS) as a modality used in equine practice, it is not described in the current manuscript. Please consider including a reference for APS and a brief paragraph where appropriate

Please considered changing the title: “1.2.1. Scaffold-based regenerative techniques”. To “ 1.2.1. Scaffold-based regenerative strategies” or in alternative “1.2.1. Scaffold-based regenerative therapies” I believe this change would be more suitable and enhance the manuscript's flow.

Line 152- TGF-β1, TGF-β2, should be explain the meaning of abbreviations. Transforming Growth Factor (TGF-β1)

Line 155- in 400x103, the number 3 should be in superscript

Line 187- As the cell source is more comprehensively addressed in lines 196 and 197, I recommend omitting the sentence 'These cells are commonly found in bone marrow (BM) and adipose tissue (AT),' and rephrasing this portion.

Line 191- They can be obtained through minimally  invasive procedures. I do not agree that bone marrow collection is not a minimally invasive procedure.

Line 212 to 216- I do not see the interest of caprine or dog model here. Please add alternatively the first work in horses.

Improve layout of Figure 4  and correct errors:

1-“ Articles dupicated and eliminated Y and N=”

2- Please consider merge two boxes and state “Articles examined and evaluated for full text eligibility n=296”

Line 389- “equine patients (Table 2).”

Table 3 – A donkey is not a Horse. Please consider changing to equine in table description.

Table 3- BMSCs. Should be BMSCs

Table 3- I do not agree the inclusion of Xing et al., 2018 in this table since is a review. Some of the information in the text can be maintained as a conclusion of this section.

Table 3- The year of the citation Villar 2015 , does not match the reference Villar 2016.

Table 4- One of the exclusion criteria was: Laboratory methodologies. Therefore I do not understand

how the work of Arévalo-Turrubiarte et al., 2022  was included in the study since it appears to be a

laboratory work.

Comments on the Quality of English Language

Minor editing of English language required

Author Response

Dear reviewer, first of all, I want to express my deepest gratitude for your review and the suggestions you have made. These suggestions have undoubtedly increased the quality of our manuscript.

The review entitled: “Regenerative medicine applied to osteoarticular diseases in equines: a systematic review” is a well written concise, systematic review that addresses a relevant topic in equine medicine. Before final approval several changes should be performed in the manuscript.

While the title refers to osteoarticular diseases in equines, the search questions were focused on musculoskeletal diseases, which is a broader topic encompassing conditions such as tendinopathies not associated to articular diseases. Therefore, the authors should either change the title to 'musculoskeletal diseases' or restructure the manuscript accordingly.

You are right, the title has been changed, to be coherent, according to your suggestion.

The simple summary primarily focuses on cell-based therapies and does not encompass all the work, particularly non-cell therapies such as PRP and ACS. Please reformulate this section accordingly.

It has been changed in the revised manuscript.

1.1. Conventional treatment of the most common osteoarticular pathologies in equine medicine”. Please review this section accordingly with the initial comment on title.

It has been changed in the revised manuscript.

Line 112 and 113: This line suggests that regenerative medicine research started recently, although the results obtained may not yet meet the desired criteria; the search was initiated years ago. Please reformulate this sentence accordingly.

It has been changed in the revised manuscript.

Line 129- While the authors acknowledge autologous protein solution (APS) as a modality used in equine practice, it is not described in the current manuscript. Please consider including a reference for APS and a brief paragraph where appropriate.

It has been added in the revised manuscript (Lines 163-168).

Please considered changing the title: “1.2.1. Scaffold-based regenerative techniques”. To “ 1.2.1. Scaffold-based regenerative strategies” or in alternative “1.2.1. Scaffold-based regenerative therapies” I believe this change would be more suitable and enhance the manuscript's flow.

It has been changed in the revised manuscript.

Line 152- TGF-β1, TGF-β2, should be explain the meaning of abbreviations. Transforming Growth Factor (TGF-β1).

It has been explained in the revised manuscript.

Line 155- in 400x103, the number 3 should be in superscript

It has been added to the revised manuscript.

Line 187- As the cell source is more comprehensively addressed in lines 196 and 197, I recommend omitting the sentence 'These cells are commonly found in bone marrow (BM) and adipose tissue (AT),' and rephrasing this portion.

It has been changed in the revised manuscript.

Line 191- They can be obtained through minimally invasive procedures. I do not agree that bone marrow collection is not a minimally invasive procedure.

It has been changed in the revised manuscript.

Line 212 to 216- I do not see the interest of caprine or dog model here. Please add alternatively the first work in horses.

It has been corrected in the revised manuscript reference 32 only refers to the treatment in horses.

Improve layout of Figure 4 and correct errors:

1-“ Articles duplicated and eliminated Y and N=”

It has been corrected in the revised manuscript.

2- Please consider merge two boxes and state “Articles examined and evaluated for full text eligibility n=296”

It has been merged.

Line 389- “equine patients (Table 2).”

It has been corrected in the revised manuscript.

Table 3 – A donkey is not a Horse. Please consider changing to equine in table description.

It has been corrected in the revised manuscript.

Table 3- BMSCs. Should be BMSCs

It has been corrected in the revised manuscript.

Table 3- I do not agree the inclusion of Xing et al., 2018 in this table since is a review. Some of the information in the text can be maintained as a conclusion of this section.

This reference has been changed in the revised manuscript.

Table 3- The year of the citation Villar 2015 , does not match the reference Villar 2016.

It has been corrected in the revised manuscript.

Table 4- One of the exclusion criteria was: Laboratory methodologies. Therefore I do not understand how the work of Arévalo-Turrubiarte et al., 2022 was included in the study since it appears to be a laboratory work.

You are right, due to the lack of information on the clinical application of MSCs-derived secretome in horse musculoskeletal diseases we had to include this in vitro study. It has been changed in the Materials and Methods section.

Reviewer 2 Report

Comments and Suggestions for Authors

the authors aimed to provide a systematic review of scientific literature in regenerative medicine in horses. the topic is highly relevant in my opinion as standardization of clinical trials and clear presentation of results is often difficult in this field.

the authors consulted different databases and main papers were retrieved. in the discussion section they discuss the different results and limit of each study, which is of high importance. Although, they dedicated a section to the limit of the study, I have some suggestion about the overall organization of the paper.

Introduction: I found it too long. Interesting and appropriate but too long. I would focus on the current therapies for musculokeletal conditions and their limits, rather than explaining their pathogenesis.

Line 1055: ..x 103? should it be 100?

paragraph 2.3 and 2.4:  I would move these lines to the results section

Tables: please order the references according to the year of publication.

Table 3 and 4: a review and an in vitro studies are added. the authors assessed that only clinical trials were chosen as eligible. please clarify

Author Response

First of all, I want to thank your kind revision that has undoubtedly improved our manuscript.

The authors aimed to provide a systematic review of scientific literature in regenerative medicine in horses. The topic is highly relevant in my opinion as standardization of clinical trials and clear presentation of results is often difficult in this field.

The authors consulted different databases and main papers were retrieved. in the discussion section they discuss the different results and limit of each study, which is of high importance. Although, they dedicated a section to the limit of the study, I have some suggestion about the overall organization of the paper.

Introduction: I found it too long. Interesting and appropriate but too long. I would focus on the current therapies for musculokeletal conditions and their limits, rather than explaining their pathogenesis.

It has been changed in the revised manuscript.

Line 1055: ..x 103? should it be 100?

It has been corrected in the revised manuscript it should be in superscript (103).

paragraph 2.3 and 2.4:  I would move these lines to the results section.

I am not sure what paragraphs you refer to.

Tables: please order the references according to the year of publication.

The references have been reordered in the revised manuscript.

Table 3 and 4: a review and an in vitro studies are added. the authors assessed that only clinical trials were chosen as eligible. please clarify.

You are right, due to the lack of information on the clinical application of MSCs-derived secretome in horse musculoskeletal diseases we had to include this in vitro study. It has been changed in the materials and methods section.

Round 2

Reviewer 2 Report

Comments and Suggestions for Authors

The authors provided most of the answers to my questions. And I find the paper more readable.

I still see the paragraphs 2.3-2.4 (line 275-293) belonging to the method section. I would move them to the results section.

At 276 the authors stated that:"This work includes articles published from 2013 to 2023. However, due to the limited knowledge of the subject, it was considered appropriate to include four articles from previous year". is this where you added information about the  review and the in vitro studies are added? I don't find this information  in the paper. it should be added.

Author Response

REVIEWER 2

Dear Sir/Madam,

Thank you very much again for your efforts to improve our manuscript. The lack he noted in the corrections he suggested in the previous review was due to our misunderstanding some points. Once we realized the significance of his suggestions, we immediately included them in the manuscript.

I still see the paragraphs 2.3-2.4 (line 275-293) belonging to the method section. I would move them to the results section.

Sorry again for our misunderstanding when you submitted your previous review of the manuscript we had in mind that the systematic review we have read, usually includes the information that you suggested changing to the Results section as part of the method used in the review. We have changed it according to your instructions because you are right, they really show data that could be considered as the starting point of the systematic review. Once we realized the meaning of your proposal, we changed these paragraphs (see lines 277-293). Thank you again for your help and we sincerely apologize for the inconvenience.

At 276 the authors stated that:"This work includes articles published from 2013 to 2023. However, due to the limited knowledge of the subject, it was considered appropriate to include four articles from previous year". is this where you added information about the  review and the in vitro studies are added? I don't find this information  in the paper. it should be added.

One of the most common tendencies in writing systematic reviews restricted to certain dates of not citing works prior to the dates under consideration is one of the main facts that we personally hate. I agree that this type of review must include, as a central point, the articles reviewed in the study. Still, we do not believe that this should restrict the use, to contextualize the information provided, of citing high-quality articles that should have the support of the results shown in the selected articles.

However, trying to balance the trend that is currently being followed, we have restricted as much as possible the use of articles prior to the date under consideration, the year 2013. Those articles reviewed by us correspond to the references:

[3] (in Table 1) Torricelli et al., 2011

[25] (in Table 2) Frisbie et al., 2007

[42] (also in table 2) Georg et al., 2010

[45] (in Table 3) Mokbel et al., 2011.

Personally we very much regret not citing and commenting on other articles that should be referenced due to their enormous quality presented on previous dates.

We reiterate our gratitude for your efforts in helping us improve the quality of the manuscript.